# Potential Applications of Computational Fluid Dynamics for Predicting Hemolysis in Mitral Paravalvular Leaks

**DOI:** 10.3390/jcm10245752

**Published:** 2021-12-09

**Authors:** Michał Kozłowski, Krzysztof Wojtas, Wojciech Orciuch, Marek Jędrzejek, Grzegorz Smolka, Wojciech Wojakowski, Łukasz Makowski

**Affiliations:** 1Department of Cardiology and Structural Heart Diseases, Medical University of Silesia, Ziołowa 47, 40-635 Katowice, Poland; jedrzejekmarek@gmail.com (M.J.); gsmolka@me.com (G.S.); wojtek.wojakowski@gmail.com (W.W.); 2Faculty of Chemical and Process Engineering, Warsaw University of Technology, Waryńskiego 1, 00-645 Warsaw, Poland; krzysztof.wojtas@pw.edu.pl (K.W.); wojciech.orciuch@pw.edu.pl (W.O.); Lukasz.Makowski.ICHiP@pw.edu.pl (Ł.M.)

**Keywords:** computational fluid dynamics, hemolysis, paravalvular leak

## Abstract

Paravalvular leaks (PVLs) may lead to hemolysis. In vitro shear stress forces above 300 Pa cause erythrocyte destruction. PVL channel dimensions may determine magnitude of shear stress forces that affect erythrocytes; however, this has not been tested. It remains unclear how different properties of PVL channels contribute to presence of hemolysis. A model of a left ventricle was created based on data from computer tomography with Slicer software PVLs of various shapes and sizes were introduced. Blood flow was simulated using ANSYS Fluent software. The following variables were examined: wall shear stress, shear stress in fluid, volume of PVL channel with shear stress exceeding 300 Pa, and duration of exposure of erythrocytes to shear stress values above 300 Pa. In all models, shear stress forces exceeded 300 Pa. Shear stress increased with blood flow rates and cross-sectional areas of any PVL. There was no linear relationship between cross-sectional area of a PVL and volume of a PVL channel with shear stress > 300 Pa. Blood flow through mitral PVLs is associated with shear stress above 300 Pa. Cross-sectional area of a PVL does not correlate with volume of a PVL channel with shear stress > 300 Pa and duration of exposure of erythrocytes to shear stress > 300 Pa.

## 1. Introduction

Paravalvular leaks (PVLs) following surgical valve replacement are relatively common. It is estimated that such pathology affects approximately 17% of patients with prosthetic aortic valves, and 22% of patients with prosthetic mitral valves at the time of discharge after surgery [1]. Several risk factors for development of PVLs have been identified: calcifications in the valve’s annulus, infective endocarditis, therapy with corticosteroids, type of valve (PVLs are more frequent in mechanical valves), and surgical technique (in the case of mitral valves, continuous sutures are associated with a greater risk than interrupted sutures). The majority of PVLs (74%) develop during the first year after surgery [2].

Hemolysis may be a serious problem in patients with PVLs. It is a common pathology in this setting—current data place hemolysis as main indication for PVL closure in 10–20% of cases. It is also possible to observe both hemolysis and heart failure symptoms due to PVL in the same patient (up to 50% of cases) [3]. Hemolysis in patients with PVLs can have various levels of severity. Benign cases can be corrected by compensatory increase of hematopoiesis in the bone marrow, with severe cases requiring repeated blood transfusions. Pathogenesis of hemolysis in patients with PVLs appears to be multifactorial, and current data indicate that several factors may be important such as: shear stress, turbulent pattern of blood flow, fluctuations of blood pressure between cardiac chambers connected by a PVL, interactions of erythrocytes with prosthetic material, and even defects of cellular membranes of erythrocytes [4,5]. One of these factors, namely shear stress distribution in the PVL channel, can currently be precisely determined using specialized computer software and principles of computational fluid dynamics (CFD). CFD is a numerical modelling method based on solving equations of differential balance of mass, momentum, and energy for a small section of fluid, known as a computational cell. This technique allows for the creation patterns of flow and assess distribution of shear stresses in various fluids, including human blood. CFD has already been used in medicine for analysis of wall shear stresses in coronary arteries and for planning of carotid endarterectomies [6,7]. In addition, CFD can be used to predict presence of PVLs in transcatheter treatment of aortic and mitral valves [8,9,10]

In this article, we present a simplified model of the left ventricle, which was created based on data obtained from a cardiac computer tomography with contrast. PVL models of various shapes and sizes were then placed around the mitral annulus, and blood flow patterns through the PVL channels were assessed.

## 2. Materials and Methods

To create a model for evaluation, a cardiac computer tomography (CCT) with contrast of a normal left ventricle was used. Anonymous CT data of a healthy left ventricle that was already present in our database was extracted. Semi-automatic segmentation of CCT data was performed in 3D Slicer software (https://www.slicer.org, accessed on 12 January 2020) [11]. A simplified 3D model of the left ventricle was created (Figure 1, Table 1), which consisted of the chamber of the left ventricle, an aortic outlet, and a mitral valve. The volume of the left ventricle was 87 mL, which was determined based on average end-diastolic and end-systolic volume from CCT. Then, PVLs of various shapes and sizes were introduced (Figure 2). Dimensions of respective PVLs are described in Table 2. All simulations were performed in systole; therefore, the aortic valve was opened, and mitral valve was closed, and backward flow towards the left atrium occurred only through the PVL channel. Intraventricular blood flow was simulated using commercial CFD software ANSYS Fluent 2020R1. The left ventricle’s numerical mesh used in simulations consisted of 500,000 polyhedral cells, with an average size of 1 mm. In the zone surrounding the channel between the left ventricle (LV) and left atrium (LA), the mesh was of higher density with an average cell size of 0.145 mm. Performed tests showed that further mesh refinement did not affect the results—velocity profiles and energy dissipation rate in the system were identical (approx. 1–2% relative difference between meshes). Blood was assumed to be a Newtonian fluid with a density of ρ = 1060 kg m^−3^ and viscosity of μ = 3.45 × 10^−3^ Pa s. Such an assumption is only acceptable for flows with shear rates above 100 s^−1^. These values are reached during blood flow through vessels of significant diameters or in the heart [12]. Geometries of PVLs that were studied are shown in Figure 2. The length of each PVL channel was 3 mm. Simulations were performed under steady-state conditions (pressure differences driving flow through the PVL channel were constant). For the purpose of this study, an 80 mmHg pressure gradient was chosen to perform all calculations, and a heart rate of 60 beats per minute was assumed. Calculations were performed for various blood volume flow rates: 147 mL/s, 240 mL/s, and 320 mL/s. The following variables were examined regarding blood flow through the PVL channel: wall shear stress, shear stress in fluid, volume of PVL channel in which wall shear stress exceeds 300 Pa, and duration of exposure of red blood cells in the PVL channel to shear stress values above 300 Pa. This study did not require the approval of an Ethics Committee.

## 3. Results

Maximal wall shear stress values (τ) and maximal shear stress in fluid ($\tau_{xy}$) for all blood flow rates are presented in Table 3.

The volume of each PVL channel in which shear stress exceeded 300 Pa, as well as duration of exposure of red blood cells to shear stress higher than 300 Pa for each PVL channel, are presented in Table 4. The smallest volume of a PVL channel in which shear stress exceeded 300 Pa was observed in the case of PVL Model I for all blood flow rates, and the highest value of this variable was noted in the case of PVL Model IV for all blood flow rates. Time of exposure of blood to shear stress above 300 Pa was comparable between PVL models for a blood flow rate of 240 mL/s, with the exception of PVL Model III, in which the shortest time was observed. Figure 3 shows relationship between CSA and maximal wall and fluid shear stresses. In addition, the spatial distribution of fluid and wall shear stresses were examined, and are presented in Figure 4 and Figure 5, respectively.

## 4. Discussion

CFD is a helpful tool in assessment of blood flow through PVL channels. It is currently possible to create models of the left ventricle, left atrium, and PVLs based on imaging data obtained from patients with real PVLs. However, for this proof-of-concept study, we decided to manually create simple models of PVL channels, since these that can be encountered in clinical scenarios may be extremely complex. Therefore, only the model of the left ventricle was based on real CT data. It is worth noting that all shapes of PVLs used in our simulations are commonly encountered in clinical practice, and for simplification purposes, we decided to utilize straight (parallel to long axis of the left ventricle) PVL channels in all models. The length of the channel was 3 mm in all models, due to several reasons. One of the most commonly used devices for PVL closure procedures, the Occlutech PLD device (Occlutech Holding, Schaffhausen Switzerland), has a double-disc design with an interconnection of exactly 3 mm. Furthermore, there is data indicating that when a PLD device is used during PVL closure procedures, higher procedural success rates are achieved in cases where the PVL channel length is less than 5 mm [13]. Therefore, we decided to choose a PVL channel length that would be desirable during PVL closure procedures. In our work, we simulated flow through PVLs for different blood volume flow rates, similar to what is observed in subjects with real PVLs. Chosen blood volume flow rates of 147 mL/s, 240 mL/s, and 320 mL/s roughly correspond to mean stroke volumes of 51 mL, 84 mL, and 112 mL, therefore we simulated various levels of left ventricular function. A total of four PVL channel models were created. Models I, II, and III represent relatively small PVLs that can be encountered clinically, with shape being the main differentiating factor—models I and II are slit-like, and model III is round/oval, with a CSA of 3.6, 5.03, and 7.9 mm^2^, respectively. Model IV represents a larger PVL, with a CSA of around 20 mm^2^. Shapes were chosen based both on our experience in PVL closure, as well as literature data [13,14]. Articles regarding PVL treatment usually employ the same nomenclature as used in our study to describe shapes of PVL channels; however, differences due to local practice may occur. This classification is based solely on visual estimation, and there are no specific mathematical definitions of various shapes. One of the key terms in discussing the risk of hemolysis in PVLs is shear stress, which is the component of stress coplanar with the material cross-section. Fluids (including blood) moving along a solid boundary will also incur shear stress at that boundary. It has been established in previous work that when shear stress exceeds 300 Pa, erythrocyte destruction (hemolysis) can be observed [4]. In our study, this critical shear stress value was exceeded in all analyzed PVL variants for all blood volume flow rates for both the wall shear stress and shear stress in fluid. Furthermore, it was observed that values of these variables were dependent on CSA of a PVL channel, but not on its circumference. Both CSA and circumference are variables that are commonly assessed with imaging during clinical diagnostics of patients with PVLs, while other properties of PVL channels, such as PVL volume or PVL wall area, are not readily available. Therefore, we also performed several derivative calculations to search for possible relationship between clinically available parameters that describe PVL channels and results obtained with CFD simulations. Values of both the maximal wall shear stress and shear stress in fluid raised rapidly in PVLs with relatively small CSA (3.6–7.9 mm^2^). Further increases in CSA resulted in relatively smaller increases in these values. These results indicate that some degree of hemolysis will be present in each PVL, since, even in channels with small CSA and low flow rates, there will be erythrocyte destruction due to shear stress forces greater than 300 Pa.

CFD also provides insight into shear stress distribution within PVL channels. Figure 4 shows that the greatest shear stress values are present at the channel entrance, while further flow towards the left atrium is free from areas where shear stress exceeds the critical value of 300 Pa.

Since we do not observe clinically significant hemolysis in every single mitral PVL, additional factors must play role in hemolysis severity. Among these, special attention should be paid to the volume of the PVL channel in which the critical shear stress value is exceeded, as well as the duration of exposure of red blood cells to the shear stress higher than 300 Pa. The relationship between values of these two parameters with levels of markers of hemolysis has never been tested in a clinical scenario; however, one could suppose that these may play a significant role. We hypothesize that, with larger volumes of PVL channels in which the critical shear stress value is exceeded, as well as longer blood residue time in such areas, more significant degree of hemolysis should be expected. It seems that there is no linear relationship between either CSA or total volume of a PVL channel, with volume of the PVL channel in which the critical shear stress value is exceeded. The greatest absolute volume of the PVL channel in which the critical shear stress value was exceeded was observed for the PVL model with the largest CSA and total volume. Interestingly, in models with relatively small CSAs (models I, II and III), we observed a decrease in volume of the PVL channel with shear stress > 300 Pa when models II and III were compared, despite the fact that model III had larger total volume. This should be interpreted along with time of blood exposure to shear stress values higher than 300 Pa. It is interesting to note that, for blood flow rate of 240 mL/s, there were only minimal differences in exposure times between PVL models I, II, and IV. Since these models differ significantly in CSAs, wall area, circumference, and total volume, it is highly unlikely that exposure times are related to these PVL channel properties.

The final property of a PVL channel that is hard to describe in mathematical terms is its shape. In clinical practice, we encounter PVLs of various shapes: crescent, oval, round, or slit-like. One way to judge shape of a PVL channel would be to report the maximal and minimal dimensions of a PVL channel. However, performing these measurements lacks standardization, and therefore they are not commonly reported. Differences in shapes may explain why, in slit-like shaped channels (models I and II), we observed an increase in PVL channel volumes with shear stress > 300 Pa, whereas in the case of model III, a decrease is noted, despite larger CSA and total volume. Due to the fact that shape seems to be an independent variable that has influence on the parameters analyzed in this study and, at the same time, is hard to define in mathematical terms, further studies are needed to empirically define these relationships.

Aside from shear stress and shape of the PVL channel, there are other factors that have to be considered when discussing the risk of hemolysis in patients with PVLs. The presence of calcifications in the PVL channel was determined to be a risk factor of hemolysis in mitral PVLs [13]. In addition, a number of genetic red blood cell membrane (RBC) defects have been distinguished that result in increased susceptibility of RBCs to hemolysis. These hereditary disorders are due to mutations in genes encoding for membrane or cytoskeletal proteins [15]. One of such diseases is hereditary spherocytosis, which is the most common non-hemolytic anemia in the Caucasian population [15]. Due to many genetic variants, this disease does not always lead to clinically detectable hemolytic anemia. However, when an additional factor that predisposes to hemolysis appears (such as a PVL), the patient may develop hemolytic anemia.

To our knowledge this was the first comprehensive use of CFD to simulate the flow of blood through PVL channels to provide detailed data regarding values of wall shear stress, shear stress in fluid, and volume of PVL channels in which shear stress exceeds 300 Pa, as well as duration of exposure of red blood cells to values of shear stress above 300 Pa.

Our study has certain limitation that must be considered. It is a theoretical study in which we created a model of the left ventricle based on real-life data. Shapes and cross-sectional areas of analyzed PVL models also resemble those commonly seen in daily practice; however, PVL channels used in this study were created manually. Unfortunately, PVL channels are rarely straight and without any bends. The tract of PVL channels can be serpiginous (C-shaped or S-shaped), and this could render our calculations inaccurate. In our scenario, all PVL models had the course of the channel perpendicular to short axis of the left ventricle. In our work, we did not take into account how rapid deceleration of blood in the left atrium can contribute to erythrocyte destruction, and such a relationship has previously been described [5]. To unequivocally determine the relationship between various parameters that describe PVL channels and clinically significant hemolysis, it would be necessary to correlate morphologies of PVL channels of various patients with this pathology with levels of markers of hemolysis such as lactate dehydrogenase, haptoglobin, free hemoglobin, and reticulocytes. Imaging data extracted from such real-life cases must be used to create individual models for CFD analysis. In our opinion, application of CFD for assessment of flow through PVL channels could help to determine the relationship between specific parameters of PVL channels that are measured with imaging (especially echocardiography) and the risk of hemolysis. This could be useful in predicting long-term results of surgical valve replacement procedures and percutaneous paravalvular leak closure interventions.

## 5. Conclusions

CFD can be used to simulate flow through mitral paravalvular leaks. In all examined PVL models, the critical value of shear stress was exceeded during blood flow, indicating that some degree of hemolysis will surely be present in this pathology. Additional properties of PVL channels, such as volume of the PVL channel in which this critical value is exceeded, as well as duration of exposure of red blood cells to shear stress > 300 Pa, may contribute to severity of hemolysis in patients. No linear relationship was found between these parameters and CSA, which is most commonly reported clinically when PVLs are described. We hypothesize that the shape of a PVL channel may be crucial in determining risk of hemolysis, but this remains to be tested in further studies.

## Figures and Tables

**Figure 1 jcm-10-05752-f001:**
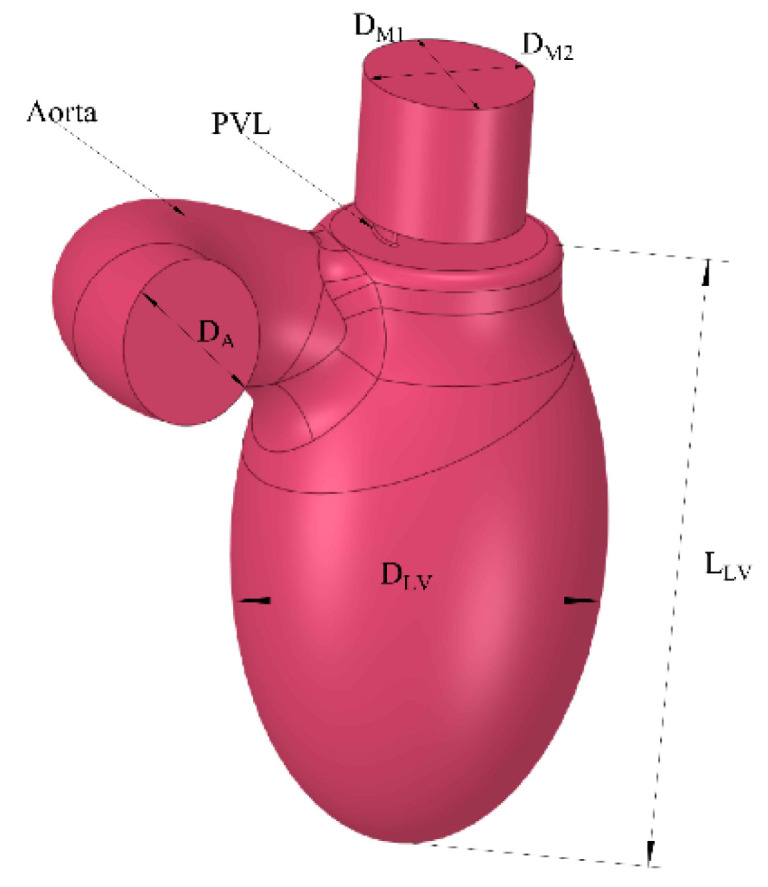
Model of the left ventricle.

**Figure 2 jcm-10-05752-f002:**
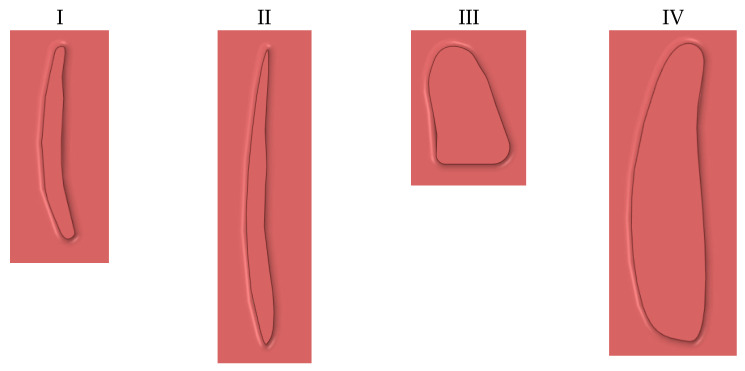
PVL geometries. I—slit-like, II—slit-like, III—oval, IV—oval.

**Figure 3 jcm-10-05752-f003:**
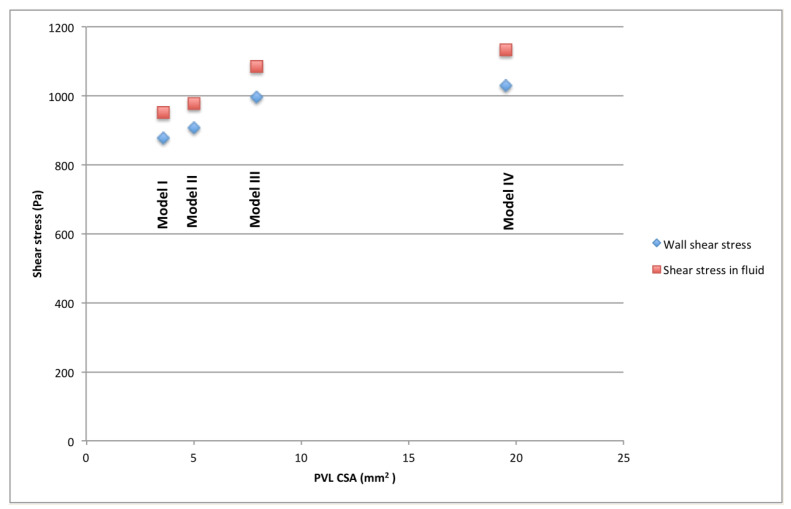
Relationship between CSA and maximal wall and fluid shear stresses.

**Figure 4 jcm-10-05752-f004:**
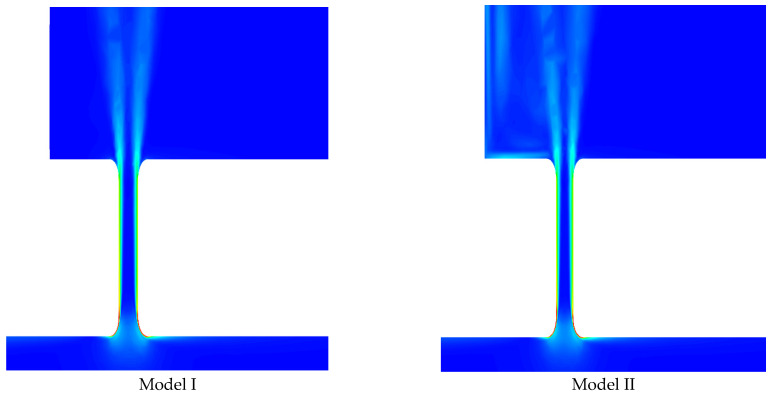
Contours of shear stress in fluid in PVL channels for each analyzed model.

**Figure 5 jcm-10-05752-f005:**
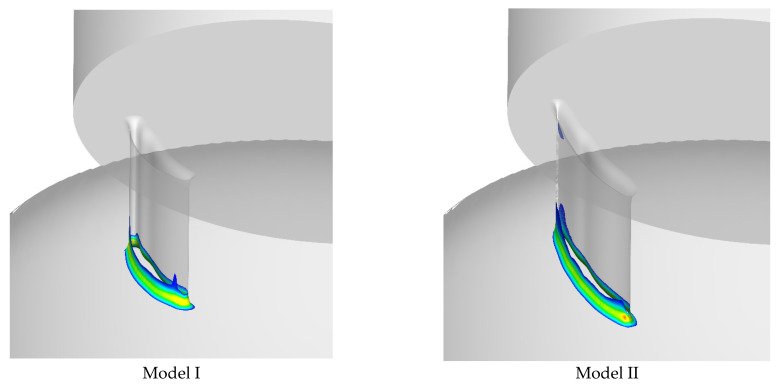
Wall shear stress distribution for each analyzed PVL model.

**Table 1 jcm-10-05752-t001:** Dimensions of the model of the left ventricle.

Dimension	Value
$D_{M1}$	24 mm
$D_{M2}$	20 mm
$D_{A}$	20 mm
$D_{LV}$	37.5 mm
$L_{LV}$	65 mm

$D_{M1}$—major diameter of mitral inflow, $D_{M2}$—minor diameter of mitral inflow, $D_{A}$—aortic diameter, $D_{LV}$—maximal diameter of the left ventricle in short axis, $L_{LV}$—maximal diameter of the left ventricle in long axis.

**Table 2 jcm-10-05752-t002:** Characteristics of PVL models.

Model Number	Cross-Sectional Area (mm^2^)	PVL Circumference (mm)	$\mathrm{PVL}\;\mathrm{Volume}\;(mm^{3})$	$\mathrm{PVL}\;\mathrm{Wall}\;\mathrm{Area}\;(mm^{2})$
I	3.6	13.6	11.5	46
II	5.03	20.1	15.8	67.5
III	7.9	11.2	24.3	69
IV	19.5	22.1	59.3	74.5

**Table 3 jcm-10-05752-t003:** Maximal wall shear stress values (τ) and maximal shear stress in fluid ($\tau_{xy}$) for all blood flow rates.

PVL Model Number	Flow (mL/s)	τ (Pa)	$\tau_{xy}$ (Pa)
I	147	845	913
	240	877	951
	320	913	992
II	147	878	947
	240	907	979
	320	941	1017
III	147	980	1070
	240	995	1085
	320	1015	1104
IV	147	1009	1113
	240	1029	1135
	320	1051	1159

**Table 4 jcm-10-05752-t004:** Volume of PVL channels with shear stress > 300 Pa and duration of exposure of red blood cells to shear stress > 300 Pa.

PVL Model Number	Flow (mL/s)	Volume (mm^3^)	Time of Exposure (μs)
I	147	0.105	8.12
	240	0.109	8.36
	320	0.118	8.84
II	147	0.153	8.71
	240	0.160	8.95
	320	0.172	9.48
III	147	0.122	4
	240	0.123	4
	320	0.128	4.78
IV	147	0.512	6.9
	240	0.637	8.53
	320	0.838	11.12

## Data Availability

The data presented in this study are contained within the article.

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
