# Peer review of "Potential Applications of Computational Fluid Dynamics for Predicting Hemolysis in Mitral Paravalvular Leaks"

_jcm, 2021, doi:10.3390/jcm10245752_

Round 1

Reviewer 1 Report

Dear authors. I have read your “Potential applications of computational fluid dynamics for predicting hemolysis in mitral paravalvular leaks” article with interest. I do congratulate you on the work performed. Most medical publications are based on the clinical data, your is a touch to the basic science with findings potentially can be applicable to the clinical practice. Despite several issues I will address below your article well-organized. It does contain all of the components I would expect for the scientific publication. All the sections more or less well-developed and described. Mostly you did a good job of synthesizing the literature available on paravalvular leak subject. I think you have answered the questions you have raised prior creating the model. Despite some limitations which are clearly understandable (difficulties to create models and calculations for serpingous shape defects) in general this is a scientifically valuable work. Mostly an article is well-written and easy to understand. Nonetheless, some minor revision for this publication is needed.

In the section Abstract

  1. You state "several shape" please describe all four different shapes and sizes. Explain why (on what basis) such shapes and sizes were used for the model.
  2. Name of the software manufacturer and place should be

In the section Introduction

  1. While describing risk factors – citation is needed.
  2. Simplify the sentence about severity of hemolysis.

In the section Methods

  1. Authors should describe in detail how did they get normal left ventricle which was used to create a model, was it from cadaver? Or they used the CT images of a healthy volunteer? If you did use a volunteer, state it and that a consent was obtained to use cardiac CT images.
  2. “3D Slicer software”, please state specifically name of the software, manufacturer and place.
  3. Authors describe that “a simplified 3D model of the left ventricle was created, which consisted of the chamber of the left ventricle, an aortic outlet and a mitral valve”. That model with left atrium missing at least in the text. From the pathophysiology point of view, in case of mitral PVL two chambers of the heart participate (left ventricle and left atrium) in hemodynamics. Readers will notice absence of left atrium in the text where you the describe the model. Authors should describe the model more accurately, explaining why in the model LA is absent?
  4. To test various scenarios of shear stress authors used “various shapes and sizes”. I suggest that authors should specify how many shapes and what sizes in the text. Also authors should explain on what basis a proposed sizes and shapes of the PVLs were used. Is it on the basis of analysis performed among the mitral PVL population? Or it is an opinion of an expert who is working in the field of mitral PVL?
  1. “All simulations were performed in systole, therefore the mitral valve was closed and backward flow occurred only through the PVL channel”. In the case of mitral PVL presence, hemodynamics a bit different. A flow of the blood occurs not only retrogradualy thru the mitral PVL, an anterograde flow into aorta is also present. Authors should explain why an anterograde flow on the model is absent. As it could influence or not the results gathered.
  2. Authors have chosen PVL channel length of 3 mm, an explanation of the chosen length is needed.

In Table 1.

  1. More detailed explanation of the variables and abbreviations is needed.

In Figure 1.

  1. PVL channels geometry also requires more detailed explanations, authors could name the shapes.

In Table 3.

  1. It is visually difficult to observe variables from the different PVL models. I suggest that the authors should separate the variables of different PVL models in four rows.

In Figure 3.

  1. Relationship between CSA and maximal wall and fluid shear stresses. Only by looking at the CSA a reader is able to understand that it is four different PVL models, authors should modify figure so that the reader can clearly and easily see which variable belongs to which PVL model.

In the section Discussion

  1. Authors discuss the causes of hemolysis occurrence, and mostly pay their attention to quantitative factors of the PVL, such as CSA, circumference, PVL volume or PVL wall area. Since the definitive mechanism for hemolysis to occur is largely unknown. In my opinion authors should broaden this part of the discussion of PVL occurrence. Probably it would be useful for the reader to know, if other factors, for example acquired or congenital “weakness” of the erythrocyte membrane or other play the role in in this mechanism.

Since not every patient develops hemolysis we do not know whether the quantitative factors of the PVL or pathology of the erythrocytes plays the key role, discus this issue as well. Additional information on this subject of “egg or chicken” will bring to the article more value, especially when the article is being published in the Journal of Clinical Medicine.

Reviewer 2 Report

This paper describes a computational flow analysis of the paravalvular leakege (PVL) likely occurring for the mitral valve under pathological conditions. results demonstrated the potential of computational flow anaylis in revealing important insights on the risk of PVL. The description of the methodology has to be improved. The following major comments need to be addressed.

page 1 line 9: please re-write this sentence. the subject of the sentence could not be "they" as this is PVL. 

page 2 line 47: CFD can be used to derive PVL in transcathetr therapy including the aortic valve and mitral valve. To support this statemednt, it is strongly recomended to add the following references. 

[1]Pasta S, Cannata S, Gentile G, Agnese V, Pilato M, Gandolfo C. Simulation of left ventricular outflow tract (LVOT) obstruction in transcatheter mitral valve-in-ring replacement. Med Eng Phys. 2020 Aug;82:40-48. doi: 10.1016/j.medengphy.2020.05.018. Epub 2020 Jul 9. PMID: 32709264.

[2] Pasta S, Cannata S, Gentile G, Di Giuseppe M, Cosentino F, Pasta F, Agnese V, Bellavia D, Raffa GM, Pilato M, Gandolfo C. Simulation study of transcatheter heart valve implantation in patients with stenotic bicuspid aortic valve. Med Biol Eng Comput. 2020 Apr;58(4):815-829. doi: 10.1007/s11517-020-02138-4. Epub 2020 Feb 6. PMID: 32026185.

[3] Pasta S, Gentile G, Raffa GM, Scardulla F, Bellavia D, Luca A, Pilato M, Scardulla C. Three-dimensional parametric modeling of bicuspid aortopathy and comparison with computational flow predictions. Artif Organs. 2017 Sep;41(9):E92-E102. doi: 10.1111/aor.12866. Epub 2017 Feb 10. PMID: 28185277.

page 2 line 61. Please justify the utilize of an ideal heart model if the CCT scan is available. Why authors did not perform a segmentation of the patient left ventricle? Please clarify this aspect.

page 2 line 59. please explain why these element size was used. have you performed mesh convergence analysis?. In general these mesh size of 1mm is large as compared to the small size of PVL areas. Please provide justification for this assumption. 

page 9 line 65. Please explain the rationale of generating such PVL shapes. are these generated from clinical data? are these ideal geometies?. if yes please highlight wye PVL areas have these shapes. 

page 2 line 72. please support the assumption of laminar flow condition here used. Please provide the reynolds number to support laminal flow condition. 

page 8 line 235. a sentence summarizing the conclusion of the present study is recomended. 

Round 2

Reviewer 2 Report

All comments and suggestions were addressed.